# Immune-Checkpoint Inhibitors (ICIs) in Metastatic Colorectal Cancer (mCRC) Patients beyond Microsatellite Instability

**DOI:** 10.3390/cancers14204974

**Published:** 2022-10-11

**Authors:** Beatrice Borelli, Carlotta Antoniotti, Martina Carullo, Marco Maria Germani, Veronica Conca, Gianluca Masi

**Affiliations:** 1Unit of Medical Oncology 2, Azienda Ospedaliero-Universitaria Pisana, 56126 Pisa, Italy; 2Department of Translational Research and New Technologies in Medicine and Surgery, University of Pisa, 56126 Pisa, Italy

**Keywords:** immune-checkpoint inhibitors, deficient mismatch repair, high microsatellite instability, *POLE* mutation, metastatic colorectal cancer

## Abstract

**Simple Summary:**

The manuscript reports the most recent literature regarding treatment with ICIs in mCRC, focusing on dMMR/MSI-H and *POLE*-mutated tumors, discussing current knowledge and potential hypotheses about molecular and/or clinical features related to intrinsic and secondary resistance to ICIs. Starting from the results of pivotal clinical trials and based on remarkable findings of CheckMate-142 and Keynote-177 trials, our manuscript provides a complete overview of the treatment with ICIs in dMMR/MSI-H metastatic colorectal cancer (mCRC), also putting an emphasis on current challenges as the optimal duration of ICIs therapy, the role of curative surgery on metastases, and we discuss mechanisms of resistance to anti-PD1 therapy in more detail. Interesting data investigating possible predictive biomarkers (e.g., TMB, PD-L1 expression, and TILs) of ICIs benefit in dMMR/MSI-H mCRC and promising ICI-based combinations able to enlarge their therapeutic efficacy are discussed.

**Abstract:**

Immune-checkpoint inhibitors (ICIs) showed impressive results in terms of activity and efficacy in metastatic colorectal cancer (mCRC) patients bearing tumors with deficient mismatch repair (dMMR) or high microsatellite instability (MSI-H). Despite that microsatellite status is the major predictive biomarker for the efficacy of ICIs, a proportion of dMMR/MSI-H mCRC tumors do not achieve benefit from immunotherapy due to the primary resistance. Deeper knowledge of biological mechanisms regulating dMMR/MSI-H CRC tumors and immune response may be useful to find new predictive biomarkers of ICIs benefit and tailor the use of immunotherapy even in dMMR/MSI-H mCRC patients. Moreover, several issues are still open, such as the secondary resection of metastases and the optimal duration of ICIs therapy in dMMR/MSI-H mCRC patients. Looking beyond microsatellite status, in a future perspective, several tools (i.e., Tumor Mutational Burden and PD-L1 expression) have been investigated to clarify their possible role as predictive biomarkers. Furthermore, a small subgroup of pMMR/MSS CRC tumors with a *POLE* mutation of the proofreading domain is characterized by hypermutated phenotype and might derive benefit from immune checkpoint inhibition. In the present work, we aim to review the most recent literature regarding treatment with ICIs in mCRC, focusing on dMMR/MSI-H and special subgroups of CRC patients. Hence, we summarize possible future targets and the most promising predictive biomarkers.

## 1. Introduction

The introduction of immune-checkpoint inhibitors (ICIs), including monoclonal antibodies against Programmed Death 1 (PD-1), Programmed Death-Ligand 1 (PD-L1), or Cytotoxic T-Lymphocyte Antigen 4 (CTLA-4), radically revolutionized the therapeutic algorithm and the prognosis of several solid tumors [1,2,3]. Nevertheless, the clinical benefit from this innovative therapeutic approach is very heterogeneous across different tumor types, likely due to the wide variability in immunophenotypes.

In the context of metastatic colorectal cancer (CRC), immune checkpoint inhibition is a recently established treatment option for the subgroup of patients bearing tumors with deficient mismatch repair or high microsatellite instability (dMMR/MSI-H) (around 5%) [4].

The biological basis of the immune sensitivity of this molecular subset of tumors relies on the high mutational burden due to the deficiency of the MMR system, leading to an increased immunogenic neoantigen load and abundant tumor infiltrating immune cells. These features, together with an immune escape via upregulation of several immune checkpoints, confer an inflamed tumor microenvironment that is highly sensitive to the immune checkpoint inhibition [5].

On the other hand, to date, no signals of efficacy of ICIs have been reported in mCRC with proficient MMR (pMMR) or microsatellite stability (MSS) [4]. Indeed, the poor or absent T-cell tumor infiltration and the reduced expression of checkpoint proteins in pMMR/MSS tumors are accountable as the major drivers of their intrinsic resistance to ICIs [6]. To this regard, with the aim of increasing the immunogenicity of these tumors and making the checkpoint blockade efficacious, several therapeutic strategies combining antitumoral agents with immunomodulatory properties (e.g., anti-VEGFR or anti-EGFR) and ICIs are under investigation [7,8,9].

Despite the high rate of durable responses and prolonged survival outcomes provided by ICIs in dMMR/MSI-H mCRC, a proportion (15–29%) of patients experiences primary ICI refractoriness or short-term clinical benefit [10,11]. Furthermore, patients who initially benefit from ICIs experience disease progression.

Herein, we propose an updated literature review of the available evidence on the use of ICIs in the therapeutic management of MSI-H/dMMR mCRC patients, and we discuss current knowledge and potential hypotheses about molecular and/or clinical features potentially related intrinsic and secondary resistance to ICIs. Finally, we focus on novel ICI-based therapeutic approaches in advanced phases of clinical research, aiming to optimize the use of this therapeutic option.

## 2. Mismatch Repair Complex and Microsatellite Status

The primary function of mismatch repair (MMR) complex is to avoid erroneous insertion and/or deletion and bases misincorporation during DNA replication. This complex is composed by the association of 5 proteins—MutS-Homologs (MSH) 2, 3, and 6; MutL-Homologue 1 (MLH1); and Post-Meiotic Segregation 2 (PMS2)—which can be combined in 3 heterodimers—MSH2-MSH6, MSH2-MSH3 and MLH1-PMS2.

The inactivation of genes encoding for MMR proteins, due to germline and/or de novo somatic mutations or to epigenetic silencing, is deleterious for genome integrity. Indeed, consequent defects in the MMR proteins impair the efficacy of the MMR complex in recognizing and repairing errors occurring during DNA replication, and finally result in the loss of genome integrity and in the accumulation of a high load of indels and frameshift mutations [12]. The high tumor mutational burden (TMB) of these cells leads to a high load of neoantigens, able to trigger a potent anti-tumor immune response by creating an immune-enriched phenotype, consisting of an abundant tumor infiltration of T-cells, which definitely are the effectors of the anti-tumor immune response [13].

A direct consequence of a defect in MMR activity is the loss of integrity of repetitive DNA genome sequences (i.e., microsatellite), consisting of mono- and di-nucleotide base pair repeats—the so-called high microsatellite instability (MSI-H) [12]. Therefore, the MSI-H phenotype is widely used as a diagnostic marker for MMR deficiency in tumor cells.

CRC tumors harboring dMMR/MSI-H have a peculiar clinical, pathological, and molecular phenotype, because they are frequently associated with right sidedness, mucinous histology, poor differentiation, distant lymph node metastases, and peritoneal carcinomatosis [14,15]. dMMR/MSI-H status is more prevalent in early-stage CRC (10–18%), as compared to the metastatic setting, where its frequency is considerably lower (about 3–5%) [16].

While the majority of dMMR/MSI-H CRCs are sporadic tumors and often caused by the epigenetic methylation of the *MLH1* gene promoter or to the acquisition of double somatic mutations in *MMR* genes, frequently associated with *BRAFV600E* mutation, only a small percentage of dMMR/MSI-H CRCs (8%) are hereditary tumors, included in the Lynch syndrome, related to germline mutations of *MMR* genes [17]. Typically, dMMR/MSI-H tumors are less sensitive to conventional chemotherapy and related to a poorer prognosis than pMMR/MSS ones [18].

## 3. ICIs in dMMR/MSI-H Metastatic Colorectal Cancer Patients

Results of pivotal clinical trials investigating ICIs in dMMR/MSI-H CRC patients are listed in Table 1. Starting from the basket trial lead by Le et al., where patients with dMMR/MSI-H chemo-refractory mCRC treated with the anti-PD-1 pembrolizumab achieved 40% of the overall response rate (ORR) compared to no objective response in those harboring pMMR/MSS tumors [4], several clinical studies investigated the role of ICIs in terms of activity and efficacy focusing on the dMMR/MSI-H subgroup.

Indeed, the activity of pembrolizumab was assessed in advanced patients with dMMR/MSI-H tumors with different histology (*N* = 86), who previously received at least one line of therapy, including CRC patients. ORR was 52% in dMMR/MSI-H CRC patients (*N* = 40) and 54% in those with dMMR/MSI-H non-CRC (*N* = 46). In the CRC cohort, the 2-year PFS rate and 2-year OS rate were 59% and 72%, respectively [23].

Subsequently, in the phase II KEYNOTE-164 study, the activity and efficacy of pembrolizumab were evaluated after ≥2 (cohort A) or ≥1 (cohort B) prior lines of therapy in patients with dMMR/MSI-H mCRC. Among 124 patients enrolled (61 in cohort A and 63 in cohort B), 44% and 30% of patients in cohort A and B, respectively, were exposed prior to 3 or more lines of therapy. After a median follow-up of 31 and 24 months, respectively, the ORR was 33% and median duration of response was not reached in both treatment cohorts. The safety profile of pembrolizumab was acceptable with 16% and 13% of treatment-related adverse events (AEs) in cohorts A and B, respectively [22].

More recently, the phase II non-comparative CheckMate-142 study provided evidence of clinical benefit of PD-1 blockade alone and in combination with CTLA-4 inhibition in previously treated patients with dMMR/MSI-high mCRC. At an extended follow-up of approximately 5 years, nivolumab monotherapy in combination with ipilimumab achieved a high response rate (ORR 39% and 65%, respectively), durable responses (median duration of response not reached in both cohorts), and relevant PFS (median PFS 13.8 and not reached, respectively) and OS (median OS 44.2 and not reached, respectively) durations, without raising long-term safety warnings [21].

Based on these remarkable findings, nivolumab and ipilimumab obtained approval from FDA and EMA as an option for patients with dMMR/MSI-H mCRC patients following prior fluoropyrimidine-based combination chemotherapy [24].

In order to challenge the hypothesis of providing a higher magnitude of the benefit from ICIs to a larger proportion of dMMR/MSI-H mCRC patients, the combination of nivolumab plus low-dose ipilimumab was investigated as a first-line treatment in a cohort of the CheckMate-142 study, including 45 patients with dMMR/MSI-high mCRC. After a median follow-up of 52.4 months, 71% of patients achieved an objective response, with a median duration of response not reached. Considerable results were reported also in terms of efficacy, with a 4-year PFS rate of 51% and OS rate of 72%. Disease progression was observed as best response in 13% of patients. Grade 3–4 of TRAEs were reported in 20% of patients, leading to discontinuation in seven cases [21].

Considering the high activity demonstrated by the dual blockade of PD-1 and CTLA4, the phase III randomized CheckMate8HW trial is currently ongoing to compare the efficacy of nivolumab alone or in combination with ipilimumab versus a standard chemotherapy as a first-line treatment in dMMR/MSI-H mCRC patients [25].

More recently, the clinical benefit of the PD-1 blockade as monotherapy in previously treated patients with dMMR/MSI-H mCRC has also been reported in the phase I/IIb multi-cohort GARNET study, testing the activity and safety of dostarlimab in prespecified advanced and recurrent solid tumors. In the cohort enrolling patients with dMMR or *POLE*-mutated non-endometrial solid tumors, 105 (56%) out of 186 patients had a refractory mCRC. In this subset of patients, dostarlimab provided a high response rate (45%) and manageable safety profile [26].

However, the most robust evidence regarding the efficacy of anti-PD1 monotherapy in dMMR/MSI-H mCRC patients are the results of the phase III KEYNOTE-177 study, which compared pembrolizumab as first-line treatment with the standard of care. In total, 307 previously untreated dMMR/MSI-H mCRC patients were randomly assigned to receive pembrolizumab for up 35 cycles or chemotherapy +/− bevacizumab or cetuximab according to investigator’s choice. In the chemotherapy arm, crossover to pembrolizumab was allowed after disease progression. The study was considered positive since one of the dual-primary endpoints (PFS and OS) was met, accordingly to pre-planned statistical hypothesis. At a median follow-up of 32.4 months, median PFS was 16.5 and 8.2 months in the pembrolizumab and chemotherapy arm, respectively (HR = 0.60, *p* = 0.0002), showing for the first time a clinical meaningful benefit for the upfront treatment with an ICI over standard treatment in dMMR/MSI-H mCRC patients [10].

In term of activity, ORR was higher in patients treated with pembrolizumab (ORR 45% vs. 33%), and the median duration of response was not reached, even though a higher percentage of patients who received pembrolizumab had disease progression (29%) as the best response compared to those in the chemotherapy arm. Efficacy results were also confirmed in terms of median progression-free survival, reaching 54 months in the pembrolizumab group compared to 24.9 months in the standard arm, highlighting the durable benefit of immunotherapy also after the first progression disease; in this regard, it should be noticed that about 60% of patients in the chemotherapy group received anti-PD1/PDL1 therapy as subsequent treatment. In terms of overall survival, pembrolizumab was not formally superior to chemotherapy; however, median OS was not reached in the pembrolizumab arm compared to 36.7 months in the chemotherapy arm (HR 0.74, *p* = 0.036), although the high crossover rate avoids the clear interpretation of OS data [26]. No unexpected toxicities were reported in the pembrolizumab arm, with grade 3 or higher AEs in 56% of patients compared to 78% in the chemotherapy group [10].

At the primary analysis, the benefit of first-line treatment with pembrolizumab over standard chemotherapy was evident across all subgroups with the exception of *KRAS/NRAS*-mutated patients [10]. However, this subgroup effect was not confirmed in the updated results; thus, the benefit of pembrolizumab seems to be consistent across the investigated subgroups [20].

Finally, results concerning KEYNOTE-177 quality of life assessment were presented. The use of pembrolizumab was associated with an improvement of health-related quality of life (HR-QoL), while a deterioration of HR-QoL scores was reported in patients treated with chemotherapy, corroborating the importance of not missing the chance of immunotherapy in this setting [27].

From a clinical perspective, the phase III randomized KEYNOTE-177 study provides the first evidence of superiority of an anti-PD1 treatment over conventional chemotherapy in dMMR/MSI-H CRC patients. Based on these results, the FDA and EMA approved pembrolizumab for the first-line treatment of patients with dMMR/MSI-H mCRC [28,29].

Frequently, dMMR/MSI-H mCRC tumors also harbor the *BRAFV600E* mutation (30%). The doublet-targeted combination with the BRAF inhibitor encorafenib plus the anti-EGFR monoclonal antibody cetuximab is today a treatment option in *BRAFV600E*-mutant mCRC patients who were previously exposed to at least one treatment regimen.

In the randomized phase III BEACON study, which included 665 mCRC patients with the *BRAFV600E* mutation, the efficacy of targeted therapy was confirmed even in the subgroup of patients with dMMR/MSI-H tumors [30]. To date, no data are available regarding the best treatment sequence (anti-PD1 inhibitors followed by anti-*BRAF* targeted therapy or vice versa) in *BRAFV600E*-mutant and dMMR/MSI-H mCRC patients. However, considering that KEYNOTE-177 study showed the efficacy of pembrolizumab as first-line treatment in dMMR/MSI-H mCRC patients regardless of *BRAF* mutation and BEACON study results [10,30] in the subgroup of *BRAFV600E*-mutant and dMMR/MSI-H patients, the first-line treatment with anti-PD1 inhibitor followed by encorafenib plus cetuximab after disease progression should be recommended. For patients with *BRAFV600E* mutation and dMMR/MSI-H, combinations of ICIs plus dual BRAF and EGFR blockade are currently under investigation. To this purpose, the phase II randomized SEAMARK study will assess the safety and efficacy of combining pembrolizumab with encorafenib plus cetuximab vs. pembrolizumab alone in patients with a previously untreated *BRAFV600E* mutation and MSI-H/dMMR mCRC [31].

Moreover, based on the fact that combining BRAF and EGFR inhibition could induce a transient MSI-H phenotype [32], Morris et al. presented fascinating results of a phase I/II trial including pre-treated patients with MSS and *BRAF V600E* mCRC who received the combination of encorafenib, cetuximab, and nivolumab [33]. The combination of targeted therapy and ICI appears to be active and well tolerated. Indeed, a phase II randomized trial to evaluate encorafenib/cetuximab with or without nivolumab in pre-treated MSS and *BRAF* V600E mCRC patients will be launched.

Although the ICIs therapy is today an established treatment option for dMMR/MSI-H mCRC, its optimal duration remains to be determined. Nevertheless, a fixed duration of 2 years or treatment until progression or unacceptable toxicity is commonly adopted. To this regard, the GERCOR NIPICOL phase II study recently evaluated one-year duration of therapy with nivolumab plus ipilimumab in previously treated dMMR/MSI-H mCRC patients. In total, 57 patients received four cycles of nivolumab 3 mg/kg plus ipilimumab 1 mg/kg, followed by nivolumab every 2 weeks until progression or a maximum of 20 cycles. The second course of nivolumab was planned for those who completed the predefined year of ICIs treatment and later experienced disease progression.

One year of treatment with nivolumab plus ipilimumab showed good and durable efficacy in terms of 3-year PFS and 3-year OS rates (70%, 95% CI 56–80 and 73%, 95% CI 58–83, respectively), comparable to those obtained by other trials in chemo-refractory patients with dMMR/MSI-H mCRC. A landmark analysis at 1 year included 42 patients who were free of disease progression and alive at 1 year. In this group, the 2-year PFS rate was 92.9%. Four patients had disease progression after the 1-year mark, suggesting that a shorter duration does not impair the efficacy of an ICI-based treatment. Furthermore, two out of three patients who were retreated with nivolumab after disease progression achieved a partial response, were alive, and were free of progression at 9 and 14 months. Based on these preliminary data, the re-treatment with the anti-PD-1 may provide additional benefit for patients experiencing late resistance after discontinuation of ICI [19].

Taken together, results from the GERCOR NIPICOL study raise the question about the need for the fixed duration of two years of therapy for all patients, which is usually recommended based on ICI therapy duration adopted in clinical trials, and for tools to identify patients requiring longer duration of therapy.

Another important unsolved issue in the management dMMR/MSI-H mCRC patients treated with ICIs is the secondary surgery on residual metastases with curative intent. In the KEYNOTE-177 study, the pathologic complete response rate was 11%. A recent analysis by Ludford et al. retrospectively assessed histopathological response in 14 dMMR/MSI-H patients who underwent surgery after ICIs treatment, both for curative (*N* = 11) and palliative (*N* = 3) intent. Pathological complete response was observed in 13 patients, despite the presence of residual tumor at the pre-operative imaging in 12 patients. The inconsistency between pre-surgical radiological and histopathological findings could be related to the tumor immune cells infiltration caused by anti-PD1/PDL1 inhibitors [34]. These results require further validation but suggest that in a proportion of dMMR/MSI-H mCRC patients treated with ICIs, curative surgery on residual radiographic tumor may be spared. Further efforts should be addressed to identify in the group of patients with positive radiological assessment those with minimal residual disease (by means of ctDNA or radiomics techniques), definitely worthy of surgical approach [35,36].

## 4. Resistance to ICIs in dMMR/MSI-H mCRC Patients

Considering the results of first-line ICIs trials (CheckMate-142 and KEYNOTE-177), it should be noticed that dMMR/MSI-H colorectal tumors likely have heterogeneous biology and, as a consequence, sensitivity to immunotherapy [10,11].

Two different categories of mCRC patients with dMMR/MSI-H can be identified: patients experiencing progressive disease as best response or a short-term clinical benefit from ICIs who are primary resistant, and those developing acquired or secondary resistance after an initial period of clinical benefit from ICIs [37].

Notably, an important number of patients (29%) treated with pembrolizumab as first-line treatment had PD as the best response [10], while only 13% were primarily refractory to nivolumab and a low dose of ipilimumab [11]. Thus, combination with ICIs, such as nivolumab plus low-dose ipilimumab, seems to partially overcome the primary resistance to anti-PD1 monotherapy in dMMR/MSI-H mCRC patients. Higher activity rate with anti-PD1 plus anti-CTLA4 compared to anti-PD1 alone were also reported in other tumor types, such as advanced melanoma [38].

Herein, we describe potential mechanisms of intrinsic resistance to ICI-based treatment in dMMR/MSI-H mCRC patients.

### 4.1. MSI Testing and Radiological Assessment

Potential mechanisms of resistance should be identified, such as the misdiagnosis of dMMR or MSI-H status and the event of radiological pseudo-progression of disease.

A post hoc analysis of a single-center cohort of 38 mCRC patients treated with ICIs revealed that primary resistance to the treatment is partially due to misdiagnosis of the MSI or dMMR status. In details, primary resistance to ICIs was observed in five patients (13%); reassessment of the MSI or MMR status revealed that three (60%) of these five resistant tumors were MSS or pMMR [39].

To avoid this issue, according to ESMO guidelines, the integrity of MMR status should be firstly assessed by means of immunohistochemical staining for expression of the four MMR proteins [37]. If immunohistochemical results are equivocal, a confirmatory analysis is mandatory. In these cases, microsatellite status should be tested by means of molecular approaches, such as polymerase chain reaction (PCR) or, more recently, Next-Generation Sequencing (NGS) panels [40].

A challenging clinical issue in the management of patients receiving ICI is the early recognition of pseudo-progression, which is a radiological phenomenon simulating an evolution of disease. The biological basis of this event relies on the massive recruitment and infiltration of T cells into the tumor, leading to a transient increase in tumor size.

The discrimination between pseudo- and true progression is crucial to avoid the discontinuation of effective treatment with ICIs or, conversely, the continuation of an effective and potentially toxic treatment. To this purpose, novel radiological criteria have been developed—immune RECIST (iRECIST) criteria—and pseudo-progression is defined as a radiological progression of lesions that is not confirmed but is followed by a sustained stability or a response to treatment [41].

A recent retrospective study involving 123 mCRC patients with dMMR/MSI-H mCRC treated with ICIs showed that after radiological review according to iRECIST criteria approximately half of patients with PD were re-classified as pseudo-progression. Globally, 10% of patients experienced pseudo-progression according to iRECIST, and none of these events occurred after the first 3 months of treatment, showing that it is an early phenomenon [42]. For these reasons, the use of iRECIST criteria in clinical trials, as well as in clinical practice, should be recommended in order to accurately estimate ORR in patients treated with immunotherapy.

### 4.2. Site of Metastatic Spread

Interestingly, even the site of metastatic spread seems to play a role in ICIs activity. A recent retrospective study showed different patterns of response in mCRC patients treated with ICIs. Notably, a limited proportion of patients experienced disease progression limited to adrenal glands. It has been showed that the use of glucocorticoids during ICIs therapy negatively impairs survival outcomes. The release of endogenous glucocorticoids by adrenal glands has been hypothesized to be able to preserve immunosuppressive cells in the tumor microenvironment, keeping this site as a sanctuary for ICIs treatment [43].

From a clinical perspective, in the case of oligo-progression involving the adrenal glands, the use stereotaxic radiotherapy or surgery on the site of progression and the prosecution of ICIs therapy might be recommended.

Moreover, Fucà et al. showed that the presence of peritoneal metastases and ascites was associated to a poor outcome in a cohort of 502 patients with dMMR/MSI-H mCRC treated with ICI [44]. This has been hypothesized to be a consequence of an immune-suppressive environment in serous cavities. In a future perspective, in the subgroup of dMMR/MSI-H mCRC patients with peritoneal involvement and ascites, novel therapeutic strategies should be investigated, as well as ICIs combination or intraperitoneal approaches.

### 4.3. Immune-Related Molecular Biomarkers

Looking at potential molecular mechanisms of intrinsic resistance to ICIs, TMB has been identified as an independent predictive biomarker of response in a retrospective cohort of dMMR/MSI-H mCRC patients treated with ICIs. The subgroup of patients with tumors with a TMB higher than 37–41 mutations/Mb was enriched of responders, as compared to that including patients with tumors with a TMB lower than 37–41 mutations/Mb [45]. As a consequence, dMMR/MSI-H CRC patients might derive a differential benefit from ICI-based approaches in relation to the TMB.

Other studies are investigating resistance mechanisms involved in disrupting antigen presentation process, such as the loss of heterozygosis of beta-2-microglobulin, which potentially affect the antigen presentation of MHC-I, inducing primary and acquired resistance to ICI [46]. Considered the crucial role of *HLA* in regulating immunity, a recent study showed that *HLA-A*03* may have a predictive role for poor response to ICI in solid tumors. Clinical outcomes of patients treated with ICI affected by different types of advanced tumors were compared to a group of patients not receiving ICI. *HLA-A*03* was found to be associated with worst outcome and absence of response only in the group patients treated with ICI [47]. Further studies, including dMMR/MSI-H CRC tumors are needed to confirm these preliminary data regarding the role of *HLA-A*03* as a potential marker of resistance to ICI.

## 5. A New Target Population for ICIs in pMMR/MSS CRC: *POLE*-Mutated Tumors

Among pMMR/MSS tumors, mutations of polymerases epsilon (*POLE*) and delta (*POLD1*) have been recently identified as potential predictive biomarkers of benefit from immunotherapy treatment.

*POLE* and *POLD1* are enzymes involved in DNA replication and have a crucial role in DNA synthesis and repair. The exonuclease domain (ED) of *POLE* and *POLD1* ensures DNA repairing in case of replication error. In case of mutations mapping in the DNA binding or catalytic site of the ED of *POLE* or *POLD1*, proofreading defects are generated. Indeed, the absent or impaired DNA repair activity leads to a ultra-mutated phenotype of tumor cells, associated with high TMB [48,49]. As a consequence, *POLE-* or *POLD1*-mutated tumors are highly immunogenic due to enrichment by mutation-associated neoantigens, and thus are potentially sensitive to immune checkpoint inhibition.

Somatic mutations of *POLE* ED could be identified in 0.5–2% of metastatic colorectal cancer, while *POLD1* ED mutations are extremely rare [50,51]. Germline mutations of *POLE* and *POLD1* ED in CRC are less frequent (0.3–0.7%), but they are involved in the development of multiple colorectal adenomas and carcinomas, causing the polymerase proofreading-associated polyposis (PPAP), which can be associated also with increased risk of non-colonic cancers [52].

A large retrospective study of 6517 patients with stage II-III CRC showed that 66 (1%) cases had *POLE* somatic mutations. Patients diagnosed with *POLE*-mutated CRC are younger at diagnosis, more frequently right-sided, and more frequently male compared to those with POLE wild-type disease. Moreover, pMMR/MSS *POLE*-mutated tumors are diagnosed in earlier stages and related to better prognosis compared to pMMR/MSS *POLE* wild-type cases [53].

From a biological perspective, *POLE*-mutated tumors displayed higher CD8+ T cell infiltration, expression of cytotoxic markers, and immune checkpoints compared with *POLE* wild-type pMMR/MSS tumors [53], while they showed a similar enrichment of CD8+ as in dMMR/MSI-H tumors, revealing a similar biology between these two subgroups of CRC tumors. Furthermore, *POLE*-mutated tumors showed a high expression of T lymphocytes markers, effector cytokines (CD8, Interferon, CXCL9, and CXCL10), and an up-regulation of immune checkpoints genes, such as PD-1, PD-L1, and CTLA4 [53].

In a small group of mCRC with *POLE*-mutated tumors (*N* = 3) and dMMR/MSI-H (*N* = 5) treated with anti-PD1, an immune-profiling analysis was performed: responders to ICIs had significantly higher densities of CD8+ PD-1+ TILs than no responders (*p* = 0.0007). On the other hand, PD-L1 expression, CD4+ T-cell density, and CD4+ FOXP3+ T-cell density did not significantly differ, but the percentage of CD4+ T-cells (Th1 phenotype) was also significantly higher in responders than no responders (*p* = 0.0007) [54]. Thus, the composition of tumor microenvironment, when enriched in CD8+, PD-1-expressing TILs, and CD4+ Th1 cells, might have a predictive role for ICIs response in *POLE*-mutated and dMMR/MSI-H CRC tumors.

Following a few pioneer case reports [55], the biological hypothesis that, in *POLE*-mutated tumors, the accumulation of mutations caused by replicative errors leads to high TMB and ultimately could result in sensitivity to ICIs, was confirmed by the promising activity of PD-1 blockade in a small cohort of pre-treated patients with MSS/pMMR advanced tumors harboring selective *POLE* pathogenic mutations in the DNA-binding or catalytic site of the exonuclease domain [56].

Among 16 patients enrolled, 6 (38%) patients achieved response, of whom 5 were mCRC patients. When activity results were analyzed according to the pathogenicity of *POLE* mutations, no response was reported in non-pathogenic subgroup, while ORR was 50% and 66% in the pathogenic and unknown significance subgroups, respectively [57].

To date, several clinical trials are currently investigating the clinical benefit of different anti-PD-L1 (durvalumab/avelumab) in *POLE*-mutated or dMMR/MSI-H CRC patients in advanced (NCT03435107, NCT03150706) [58,59] and early stage patients (NCT03827044) [60]. Finally, in the randomized POLEM trial, the addition of avelumab to standard adjuvant fluoropyrimidine-based chemotherapy will be assessed in stage III resected dMMR/MSI-H or *POLE*-mutant CRC patients [60].

## 6. Potential Predictive Biomarkers of ICIs Benefit among dMMR/MSI-H mCRC Patients

To date, dMMR or MSI-H status is an evidence-based tissue-agnostic biomarker predictive of benefit from ICIs. However, several biomarkers other than dMMR/MSI-H are under investigation as positive predictors of benefit from ICI.

### 6.1. Tumor Mutational Burden (TMB)

TMB, defined as the number of mutations per coding area of tumor genome, is considered one of the most promising biomarkers. Recently, the FDA approved the tissue-agnostic use of pembrolizumab in treatment-refractory solid tumors with a TMB of 10 or more mut/Mb [61].

In the mCRC scenario, the role of TMB has not been established yet, as well as an optimal mCRC-specific TMB cut-off able to predict the benefit from ICI-based strategies.

To this regard, in a retrospective series, among 22 MSI-H mCRC patients treated with anti-PD-1/PD-L1, the optimal predictive cut-point for TMB was estimated between 37 and 41 mutations/Mb. Patients with tumors with higher TMB achieved higher response rates and longer survival as compared to those with tumors with lower TMB, thus suggesting that TMB may serve as an addition predictive biomarker within the current standard dMMR/MSI-H [45].

Conversely, more recent data clearly showed that the TMB universal cut-point of 10 mut/Mb does not predict the benefit from an ICI-based treatment in MSS/pMMR mCRC tumors [62].

Recently, Salem et al. analyzed TMB and the MMR protein heterodimer mutation in a large cohort of MSI-H solid tumors. Colorectal cancers with MLH1/PMS2 mutations had higher TMB than endometrial cancer and other tumor histologies, showing that the TMB count is widely variable according to both type of MMR defect and tumor site origin. Based on these data, TMB heterogeneity within dMMR/MSI-H tumors may contribute to explain the spectrum of benefit from checkpoint blockade [63].

Taken together, data about TMB in mCRC highlighted that the predictive value of a single universal cut-point to define high TMB associated with benefit from ICI-based therapies, both within dMMR/MSI-H and pMMR/MSS mCRC tumors, would be improved with tumor-specific thresholds.

### 6.2. PD-L1 Expression and TILs

Moving forward among possible predictive biomarkers, PD-L1 expression is useful to predict benefit from ICIs in some solid tumors, such as NSCLC. However, PD-L1 expression in mCRC is not associated to immunotherapy response. In a retrospective study, PD-1 and PD-L1 expression in pMMR/MSS and dMMR/MSI-H CRC patients was investigated in term of prognostic relevance. Among 389 CRC patients, 68 dMMR/MSI-H tumors were related to high PD-1 and PD-L1 expression by both staining intensity and the percentage of cell staining. In pMMR/MSS tumors, PD-1 and PD-L1 expression were not associated with relapse-free survival (RFS), while in MSI-H tumors, high expression of PD-L1 was associated with worse RFS compared to those with low PD-L1 expression. At the multivariate analysis, both PD-1 and PD-L1 expression confirmed their independent association with recurrence-free survival only in the dMMR/MSI-H group, while they did not in the pMMR/MSS group [64,65]. These results were consistent with the biological role of PD-L1, since its expression on tumor cells and the binding with PD-1 inactivates T-cells and allows for tumor cells to potentially evade immune surveillance.

Moreover, PD-L1 expression is a dynamic biomarker and may be induced under the pression of therapeutic agents, such as chemo- or target therapy. Despite its important predictive role in other gastrointestinal tumors (i.e., EJG cancer) with regard to the use of ICIs, no evidence is available for the use of PD-L1 expression as predictive biomarker in dMMR/MSI-H mCRC patients.

Recently, the role of TILs in dMMR/MSI-H tumors has been investigated. TILs and TMB of 85 dMMR/MSI-H samples were performed, and patients were divided in responders or refractory to ICIs. Higher RR were observed in patients with high number of TILs compared to those with low number of TILs (ORR: 71% vs. 43%). Significantly higher PFS and OS were reported in the TILs-high group. In summary, considering the entire body of the evidence TILs are promising biomarkers of ICIs efficacy in dMMR/MSI-H mCRC and might be implemented in future clinical trials [66].

### 6.3. Gut Microbiota

In the last years, growing evidence revealed a possible role of gut microbiota in CRC carcinogenesis. Several factors, such as diet, antibiotics, and immune system, as well as cancer, are involved in temporary or permanent alteration of intestinal microbiota [67]. It has also been proved that gut microbiota is essential for the maturation of intestinal immune system. Intestinal bacteria interacting with normal colonic mucosa and gut dysbiosis may promote tumor progression through the loss of protective bacterial populations and the enrichment of microbial communities, such as *Fusobacterium nucleatum* (FN), enterotoxigenic *Bacteroides fragilis* (ETBF), and *Peptostreptococcus anaerobius*. Furthermore, it has been reported that, in non-CRC tumors treated with ICIs, the use of antibiotics may change the quantity and functions of colonic immune cells, and the composition of gut microbiota might negatively impair treatment effectiveness [68,69].

The composition of gut microbiota is also able to enhance intestinal immune response: the Bifidobacterium, both alone and in combination with anti-PD-L1 immunotherapy, modulates dendritic cell function and improves the anti-tumor activity of CD8+ T cells [70]. Several studies demonstrated that FN is implicated in CRC pathogenesis, and it seems to inhibit antitumor immune response against CRC cells. According to recent studies, FN DNA is significantly enriched in dMMR/MSI-H CRC tumors [71,72]. Furthermore, FN seem to favor the evolution of serrated polyps, which are precursors of sporadic dMMR/MSI-H or CIMP-H CRCs [73,74].

Notably, FN DNA sequence has been assessed in 160 dMMR/MSI-H CRC tumor samples, which were classified in FN-high, FN-low, or FN-negative. A higher density of CD68+ tumor-infiltrating macrophages and promoter CpG island hypermethylation of the *CDKN2A* gene were found in FN-high samples, which showed an immune-enriched TME compared to FN-low/negative tumors [75]. In a future perspective, deeper knowledge in understanding the biological mechanisms underlying gut microbiota, CRC, and tumor response to ICIs will allow us to recommend or avoid some antibiotics categories during ICIs treatment in order to obtain an immune-responsive TME.

## 7. Overcoming Resistance to ICIs in dMMR/MSI-H CRC: Future Perspectives

In the recent years, deeper knowledge about the mutational landscape of dMMR/MSI-H CRC has been crucial to find promising biomarkers and understanding the complex mechanisms underpinning primary and secondary resistance to ICIs.

Several immune checkpoint ligands expressed by T cells may inhibits immune response in dMMR/MSI-H CRC. For instance, CD274 is an immune checkpoint protein that suppress T-cell function, promoting immune evasion. A similar immunosuppressive role was reported for anti-lymphocyte activating antigen-3 (LAG-3) and indoleamine 2,3-dioxygenase 1 (IDO1), which are expressed both in CRC cells and T cells [76]. As a consequence, targeting these immune checkpoint ligands with their corresponding inhibitors might synergistically act with anti-PD1/PDL1 drugs, improving immune response against cancer cells.

However, the complexity of biological mechanisms of resistance to ICIs in dMMR/MSI-H CRC are still not clear. It has been reported that inactivating mutations of *JAK1* or *JAK2* are involved in both primary and secondary resistance to anti-PD1 inhibitors in solid tumors [77]. JAK family kinases mediates signaling of the interferon (IFN), IL-6, and IL-2, regulating transcription factors and cell proliferation. *JAK1* frameshift mutationscan be related to a pan-cancer adaptation to the immune response that occurs against highly mutated MSI-H tumors [78].

Finally, several immune-escape and immunosuppressive mechanisms are implicated in resistance to ICIs in solid tumors [79]. Furthermore, the optimal therapeutic choice after progression to ICI is still not defined.

In Table 2, the most promising immune targets and their biological functions are displayed. Several ICIs in combination with drugs against immunomodulatory molecules are under investigation in dMMR/MSI-H colorectal and no colorectal tumors with the aim to enlarge the efficacy of immunotherapy and overcame primary resistance.

In non-colorectal tumor type, ongoing strategies are evaluating the addition of monoclonal antibodies, including anti-lymphocyte activating antigen-3 (LAG-3), in combination to nivolumab in dMMR patients who previous progressed to anti-PD1 therapy. LAG-3, as PD1, plays an immunomodulatory role, binding MCH class II and inducing the activation of Tregs and suppression of CD8+ T-cells and DC [80]. LAG-3 and PD-1 synergistically regulate T cells, promoting immune escape; thus, dual blockage should activate anti-tumor immunity, and this strategy could be efficacious to overcome primary resistance to ICIs in dMMR/MSI-H CRC patients. In this scenario, we should also mention the immunoregulatory role of TIM-3, which inhibits Th1 cells response. Preclinical models of NSCLC showed that TIM-3 was upregulated in TIL after progression to an anti-PD1 inhibitor [81].

A recent phase Ib study assessed the activity and safety of anti-PD1 with or without anti-TIM3 antibody in patients with dMMR/MSI-H advanced solid tumors. The combination with anti-TIM3 antibody did not provide significant activity rate (ORR: 5%) in the cohort of patients previously treated with anti-PD1/PDL1, where most patients had dMMR/MSI-H CRC (17 out of 22 patients in the entire cohort). While anti-PD1 alone and the combination with anti-TIM3 showed considerable clinical activity (ORR: 33% and 45%, respectively) in dMMR/MSI-H solid tumors naive to prior PD-1/PD-L1 inhibitor therapy [82]. Thus, anti-TIM3 plus anti-PD1 combination may require further investigation in patients with dMMR/MSI-H not previously exposed to ICIs.

Another possible immune target could be the T cell immunoreceptor with immunoglobulin and ITIM domain (TIGIT), which regulates T cell-mediated and natural killer cell-mediated tumor recognition in vivo and in vitro. Dual PD-1/TIGIT blockade potently increases tumor antigen-specific CD8+ T cell expansion and function in vitro and promotes tumor rejection in mouse tumor models. 

A phase II multicohort trial has been recently designed and is currently ongoing in order to improve PD-1 blockage adding different inhibition targets in dMMR/MSI-H mCRC patients. This study will evaluate the activity of anti-PD1 alone or in combination with anti-CTLA4, anti-LAG-3, anti-TIGIT, or anti-ILT4, respectively [83].

Nowadays, immunosuppressive effects of vascular endothelial growth factor (VEGF) are well recognized. Indeed, VEGF inhibits various immune cells present in the tumor microenvironment, contributes to tumor-associated immune deficiency, and reduces the antigen presentation process by DCs [84,85]. As a consequence, inhibiting VEGF-mediated immunosuppressive effects, through anti-VEGF monoclonal antibody, such as bevacizumab, T cells in the TME might increase immune response against cancer cells and synergize with ICIs activity. Moreover, the addition of chemotherapy might be related to increase neoantigens release, promoting the activity of immune response [86].

On these bases, the ongoing phase III COMMIT study will evaluate whether chemo-immunotherapy with FOLFOX/bevacizumab plus atezolizumab will provide an additional clinical benefit over atezolizumab alone as first-line therapy in microsatellite instable mCRC patients [87]. Furthermore, a phase II study is currently ongoing in order to assess the activity of bevacizumab plus atezolizumab in chemo-resistant dMMR/MSI-H patients [88].

In conclusion, biological mechanisms of dMMR/MSI-H CRC tumors are currently under investigation, and deeper knowledge about ICIs resistance pathways are urgently needed to better refine candidates to receive anti-PD1 +/− anti-CTLA4 treatment.

## 8. Conclusions

In conclusion, ICIs are the new standard of care in the first-line treatment, and they definitely improved the prognosis of dMMR/MSI-H mCRC patients. Many questions are still open, such as mechanisms underlying primary resistance in dMMR/MSI-H CRC tumors, the optimal duration of ICIs therapy, and the role of secondary resection of metastases after anti-PD1 treatment. In the near future, robust results regarding the activity of immunotherapy in *POLE*-mutated pMMR/MSS CRC patients are awaited. The role of several predictive biomarkers of ICIs benefit is currently under investigation (e.g., TMB and TILs) in mCRC patients, but nowadays, none of them seem to be helpful in refining patients selection. To date, the research of predictive biomarkers and the optimal treatment strategy after ICIs progression will be the next future challenge in dMMR/MSI-H mCRC patients.

## Figures and Tables

**Table 1 cancers-14-04974-t001:** Clinical trials with ICIs in dMMR/MSI-H mCRC patients.

Study Name	Phase	Line of Treatment	*N* Patients	Agent(s)	Median Follow-Up	Results
GERCOR NIPICOL [19]	II	≥1st line	*N* = 57	Nivolumab plus Ipilimumab ^#^ (max 20 cycles)	34.5 mo	3-yrs PFS rate: 70%
						3-yrs OS rate: 73%
KEYNOTE-177 [10,20]	III	1st line	*N* = 307	Pembrolizumab vs. Chemotherapy plus target agents	44.5 mo	mPFS: 16.5 vs. 8.2 mo, HR = 0.59, *p* = 0.0008
						mOS: NR vs. 36.7 mo, HR = 0.74; p = 0.036
						[OS superiority not demonstrated]
						3-yrs PFS rate: 42% vs. 11%
						ORR: 45% vs. 33%
						DCR: 65% vs. 75%
Checkmate-142 [21]	II	1st line	*N* = 45	Nivolumab plus low-dose Ipilimumab ^##^	52 mo	4-yrs PFS rate: 51%
						4-yrs OS rate: 72%
						ORR: 71%
						DCR: 84%
		≥2nd line	*N* = 119	Nivolumab plus low-dose Ipilimumab ^#^	64 mo	4-yrs PFS rate: 54%
						4-yrs OS rate: 71%
						ORR: 65%
						DCR: 81%
		≥2nd line	*N* = 74	Nivolumab	70 mo	4-yrs PFS rate: 36%
						4-yrs OS rate: 49%
						ORR: 39%
						DCR: 69%
KEYNOTE-164 [22]	II	cohort A, ≥3rd line	*N* = 61	Pembrolizumab	31 mo	12-mo OS rate: 72%
						ORR: 33%
						DCR: 51%
		cohort B, ≥2nd line	*N*= 63	Pembrolizumab	24 mo	12-mo PFS rate: 41%
						12-mo OS rate: 76%
						ORR: 33%
						DCR: 57%
KEYNOTE-016 [4]	II	≥3rd line	*N* = 11	Pembrolizumab	36 weeks	20-weeks PFS rate: 78%
						ORR: 40%
						DCR: 90%

Abbreviations: ORR, overall response rate; DCR, disease control rate; PFS, progression-free survival; OS, overall survival; max, maximum; yrs, years; mo, months. ^#^ Nivolumab 3 mg/kg + Ipilimumab 1 mg/kg every 3 weeks (4 doses) followed by Nivolumab 3 mg/kg biweekly. ^##^ Nivolumab 3 mg/kg biweekly + Ipi 1 mg/kg every 6 weeks.

**Table 2 cancers-14-04974-t002:** Potential therapeutic targets in combination with anti-PD1/PDL1.

Target	Site of Expression	Biological Role
CD274	APC	Reduce T-CD4+ function
LAG-3	T-cells	Activation of Tregs and suppression of CD8+ T-cells and DC
IDO1	T-cells DC Macrophages	Catabolic enzyme involved in the degradation of tryptophan. Increase immunosuppressive cells, such as Treg
TIM-3	T-cells	Inhibition of Th1 cell response
TIGIT	T-cells NK cells	Decrease of cytokine production and T-effector functions

Abbreviations: LAG-3: anti-lymphocyte activating antigen-3; IDO1: Indoleamine 2, 3-dioxygenase; TIM-3: T cell immunoglobulin-3; TIGIT: T cell immunoreceptor with immunoglobulin and ITIM domain; APC: antigen-presenting cell; DC: dendritic cells; NK cells: natural killer cell.

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
