# Peer review of "Immune-Checkpoint Inhibitors (ICIs) in Metastatic Colorectal Cancer (mCRC) Patients beyond Microsatellite Instability"

_cancers, 2022, doi:10.3390/cancers14204974_

Round 1
Reviewer 1 Report
This is a review article that discussed the efficacy and limitations of immune checkpoint inhibitors (ICIs) in dMMR/MSI-H metastatic colorectal cancer (mCRC). It also touched the recent evidence that supports the use of ICIs in POLE-mutant tumors. In brief, this is a good review article but based on the simple summary and abstract, I thought there would be more discussion on resistance mechanisms and novel biomarkers. Please consider addressing the following comments.
- page 2 line 54 "immune tumour infiltration": tumor infiltrating immune cells may be more accurate
-page 2 line 55 "immune escape via upregulation..." needs to be backed up by appropriate literature
-page 5 discussion on BRAF mutant mCRC: may be worthwhile to mention the recent ASCO abstract on the combination of nivolumab, cetuximab and encorafenib in BRAF mutant, pMMR mCRC
-page 5 discussion on NIPICOL study: this study looked at a fixed duration of 1 year of therapy, why it raised the question on the "need for the fixed duration of TWO years of therapy"
-The section on "resistance to ICIs in a dMMR/MSI-H mCRC patients" seems quite limited. The only subsection 4.1 discussed about MSI testing and pseudoprogression. It did not discuss other primary or secondary resistance mechanisms. Perhaps, some of the contents were accidentally removed?
Author Response
-
Point 1: Page 2 line 54 "immune tumour infiltration": tumor infiltrating immune cells may be more accurate.
Following your suggestion, we modified this statement in our manuscript.
-
Point 2: Page 2 line 55 "immune escape via upregulation..." needs to be backed up by appropriate literature.
We added appropriate literature to this regard.
-
Point 3: page 5 discussion on BRAF mutant mCRC: may be worthwhile to mention the recent ASCO abstract on the combination of nivolumab, cetuximab and encorafenib in BRAF mutant, pMMR mCRC.
We appreciated your suggestion and we implemented our manuscript mentioning this work.
-
Point 4: Page 5 discussion on NIPICOL study: this study looked at a fixed duration of 1 year of therapy, why it raised the question on the "need for the fixed duration of TWO years of therapy".
We stated “Taken together, results from the GERCOR NIPICOL study raise the question about the need for the fixed duration of two years of therapy for all patients and for tools to identify patients requiring longer duration of therapy”. Our question raised from the fact that optimal duration of 2 years or treatment until progression or unacceptable toxicity is commonly adopted based on the results of randomized trials investigating ICI treatment in dMMR/MSI-H mCRC patients. We added an explanation in our manuscript.
-
Point 5: The section on "resistance to ICIs in a dMMR/MSI-H mCRC patients" seems quite limited. The only subsection 4.1 discussed about MSI testing and pseudoprogression. It did not discuss other primary or secondary resistance mechanisms. Perhaps, some of the contents were accidentally removed?
In our manuscript we discussed only widely validate factors of ICI resistance. However, we appreciate this comment and we implemented this section with other possible mechanisms of resistance to ICI.

Reviewer 2 Report
Patients with metastatic colorectal cancer with deficient mismatch repair and high microsatellite instability (dMMR/MSI-H) receive little benefit from chemo or chemoradiotherapy.
Results from several clinical trials have reported dMMR/MSI-H as an indicator of immunotherapy efficacy, suggesting the potential use of immune checkpoint inhibitors (ICI) for this subset of diseases.
In this article, Borelli and colleagues discuss the diagnostic and prognostic characteristics, a potential reason for resistance in patients with metastatic colorectal cancer with dMMR/MSI-H.
The assessment of dMMR/MSI-H is recently approved by the FDA and EMA as a parameter to define the optimal candidate for ICI therapy.
This is a well written review on a very interesting and up-to-date topic
Author Response
We are very pleased about your comment and we would like to thank you for appreciating our work.
Please see also the attachment.

Reviewer 3 Report
The manuscript by Borelli et al reviews treatments with immune-checkpoint inhibitors in colorectal cancer (CRC) while focusing on deficient mismatch repair or high microsatellite instability and POLE-mutated metastatic colorectal cancers.
Resistance and approach to overcome resistance to immune-checkpoint inhibitors in dMMR/MSI-H mCRC patients are discussed and the authors suggest that POLE mutated tumors could be a new target for ICIs in proficient MMR (pMMR) or microsatellite stability (MSS) CRC.
The abstract and summary are lucid and conclusions are well-thought out.
Clarity and context in this paper are good.
References are comprehensive.
No major suggestions for improvements.
Minor language editing is needed.
Author Response
We are very pleased about your comment and we would like to thank you for appreciating our work. We revised our language to improve the quality of this manuscript. We wanted to thank you for your kind recommendation.
Please see the attachment.

Round 2
Reviewer 1 Report
- Section 4 on resistance to ICIs is not well organized. The subheading of 4.1 seems unnecessary.
- Please consider including this reference, Zhuo N, Liu C, Zhang Q, Li J, Zhang X, Gong J, Lu M, Peng Z, Zhou J, Wang X, Jiao X, Wang Y, Wang Y, Gao M, Shen L, Lu Z. Characteristics and Prognosis of Acquired Resistance to Immune Checkpoint Inhibitors in Gastrointestinal Cancer. JAMA Netw Open. 2022 Mar 1;5(3):e224637. doi: 10.1001/jamanetworkopen.2022.4637. PMID: 35348710; PMCID: PMC8965636. It appears that dMMR CRC has a higher propensity for peritoneal recurrence compared to pMMR CRC.
Author Response
- Section 4 on resistance to ICIs is not well organized. The subheading of 4.1 seems unnecessary.
We’ve reorganised section 4 by adding different subheadings. Meanwhile, we revised our language to improve the quality of this manuscript. We wanted to thank you for your kind recommendation.
- Please consider including this reference, Zhuo N, Liu C, Zhang Q, Li J, Zhang X, Gong J, Lu M, Peng Z, Zhou J, Wang X, Jiao X, Wang Y, Wang Y, Gao M, Shen L, Lu Z. Characteristics and Prognosis of Acquired Resistance to Immune Checkpoint Inhibitors in Gastrointestinal Cancer. JAMA Netw Open. 2022 Mar 1;5(3):e224637. doi: 10.1001/jamanetworkopen.2022.4637. PMID: 35348710; PMCID: PMC8965636. It appears that dMMR CRC has a higher propensity for peritoneal recurrence compared to pMMR CRC.
We've included this reference in our manuscript. We thank you for pointing this out.
